# Microbial Bioremediation and Biodegradation of Petroleum Products—A Mini Review

Jeremiah A. Adedeji [1,*], Emmanuel Kweinor Tetteh [2], Mark Opoku Amankwa [3], Dennis Asante-Sackey [4], Samuel Ofori-Frimpong [5], Edward Kwaku Armah [6], Sudesh Rathilal [2], Amir H. Mohammadi [1] and Maggie Chetty [2]

1   Discipline of Chemical Engineering, School of Engineering, University of KwaZulu-Natal, Howard College Campus, King George V Avenue, Durban 4041, South Africa
2   Green Engineering Research Group, Faculty of Engineering and the Built Environment, Department of Chemical Engineering, Durban University of Technology, Steve Biko Campus Block S4 Level 1, Box 1334, Durban 4000, South Africa
3   Banda Singh Bahadur Hostel, Punjabi University, International Wing, Patiala 147002, India
4   Department of Chemical and Biomedical Engineering, FAMU-FSU College of Engineering, 2525 Pottsdamer St., Tallahassee, FL 32310, USA
5   Faculty of Accounting and Informatics, Department of Information Technology, Durban University of Technology, Ritson Campus, Durban 4000, South Africa
6   Department of Applied Chemistry and Biochemistry, School of Chemical and Biochemical Sciences, C.K. Tedam University of Technology and Applied Sciences, Navrongo P.O. Box 24, Ghana
*   Correspondence: jerry_4real@live.com

**Abstract:** The demand for technological and industrial change has become heavily dependent on the availability and use of petroleum products as a source of energy for socio-economic development. Notwithstanding, petroleum and petrochemical products are strongly related to global economic activities, and their extensive distribution, refining processes, and final routes into the environment pose a threat to human health and the ecosystem. Additional global environmental challenges related to the toxicological impact of air, soil, and water pollutants from hydrocarbons are carcinogenic to animals and humans. Therefore, it is practical to introduce biodegradation as a biological catalyst to address the remediation of petroleum-contaminated ecosystems, adverse impacts, the complexity of hydrocarbons, and resistance to biodegradation. This review presents the bioremediation of petroleum hydrocarbon contaminants in water and soil, focusing on petroleum biodegradable microorganisms essential for the biodegradation of petroleum contaminants. Moreover, explore the mineralization and transformation of complex organic and inorganic contaminants into other simpler compounds by biological agents. In addition, physicochemical and biological factors affecting biodegradation mechanisms and enzymatic systems are expanded. Finally, recent studies on bioremediation techniques with economic prospects for petroleum spill remediation are highlighted.

**Keywords:** bioremediation; biodegradation; petroleum products; hydrocarbons; microorganism; phytoremediation

## 1. Introduction

Crude oil is a critical and important commodity that currently dominates the global market [1]. The chemical makeup of petroleum includes a complex combination of aromatic hydrocarbons, aliphatic hydrocarbons, heterocyclic hydrocarbons, asphaltenes, and non-hydrocarbon compounds. Around 60–90% of this is classified as biodegradable [2]. In recent decades, severe environmental pollution and associated deficiencies have been caused by advances in the petroleum industry in the extraction, haulage, and storage of petroleum products in underground reservoirs, including refining processes [2]. Total petroleum hydrocarbons (TPHs) and fuels such as gasoline, diesel, kerosene, and lubricating oils or greases are typically hydrocarbon substances derived from petroleum sources.

Soil pollution from petroleum is also one of the serious global problems. Everyday operations such as oil exploration, waste disposal (disposal of fuel and oil), and accidental spills cause serious environmental problems that lead to oxidative stress and alter the chemical composition of soils with low nutrient availability [3]. Petroleum has a negative impact on seed germination, reduces photosynthetic pigments, slows absorption, inhibits root growth, causes leaf defects, and causes cellular damage. Others include disruption of biological membranes, disruption of signaling of metabolic routes, and disruption of plant root structure [4]. Studies have shown that low molecular weight hydrocarbons can enter plant cells and destroy their plant. However, this occurs when the development of cancer and other diseases is associated with petroleum and its derivatives, as there is evidence of petroleum contamination in diagnosing nervous system depression, anesthesia, and eye irritation in humans [5]. Due to the high toxicity, carcinogenicity, mutagenicity, and teratogenicity of petroleum pollutants, their accumulation rate predominantly affects the entire human food chain. This indicates that petroleum contamination does not negatively affect plant growth but also affects people and the environment. The term biodegradation has been defined as the biologically catalyzed reduction in the complexity of chemical compounds [6]. Organic substances such as these petroleum contaminants or products mentioned above are reduced into minor compounds by living microbes, reducing their defects. The environmental degradation of organic petroleum and other aromatic compounds is a complex task. The quantitative and qualitative characteristics are largely determined by the type and quantity of oil or petroleum products used. Others include seasonal environmental conditions, environmental conditions, and microbial community composition [7]. The activity of the microbial population impacts a significant role in the biodegradation of pollutants from petroleum hydrocarbons and petrochemicals have been extensively studied, although growth rates vary [8]. These microbes arise from bacteria, fungi, yeast, and some algae that have been found to break down hydrocarbons in motor oil [9]. Studies over the last few decades have also shown that there is only sparse information on the role of algae and protozoa in their performance in the biodegradation of petroleum products [9]. The importance of these microorganisms in the biodegradation process arises from the microbial conversion of organic pollutants, which usually occurs because the organisms can use the pollutants for their energy needs, growth, and reproduction [6]. This is predominantly an adaptive process, usually dictated by environmental conditions. The biodegradation process varies widely, with carbon dioxide reported as the end product of the degradation process [5,6]. To convert or yield a wide range of compounds, including hydrocarbons, polychlorinated biphenyls (PCBs), polyaromatic hydrocarbons (PAHs), radionuclides, and metals, some microorganisms have a surprising, naturally occurring catabolic diversity. Genetic potentials and environmental parameters such as temperature, oxygen, and quantity of nutrients, i.e., nitrogen and phosphorus, moisture, and pH, have been found to control the degree and effectiveness of degradation in soil/water [10]. Recent reviews have either focused on the metabolism of the hydrocarbon degradation process, factors influencing the process, the co-combination of various biological techniques, and some specific petroleum products. In this review, the bioremediation of petroleum hydrocarbon pollutants in water and soil, the role of microorganisms in the biodegradation of petroleum products, and the conversion of complex organic and inorganic pollutants into other simpler compounds by biological agents were discussed with recent applications. Moreover, the physicochemical and biological factors influencing the biodegradation mechanism and the enzymatic systems are highlighted along with bioremediation techniques.

## 2. Petroleum Products

The term petroleum was derived from the words petra (stone) and oleum (oil), which occurs naturally and is generally processed into various products by the refinery [11]. Crude oil is known as a petroleum hydrocarbon, generally composed of hydrocarbons, heteroatom compounds, and relatively small concentrations of metallic components. The aliphatics, aromatics, asphaltenes, and resins could be classified as general classes of

petroleum derivatives. The aliphatic fraction consists of straight-chain alkanes, branched alkanes (isoalkanes), and cycloalkanes (naphthenes). The aromatic fraction is the volatile monoaromatic hydrocarbons such as benzene, toluene, and xylenes; polycyclic aromatic hydrocarbons (PAH); naphthenic aromatics; and aromatic sulfur compounds such as thiophenes and dibenzothiophenes. The asphaltene (ketones, phenols, fatty acids, esters, and porphyrins) and resin (sulfoxides, amides, pyridines, quinolines, and carbazoles) parts, which are composed of polar molecules with N, S, and $O_2$ [12]. These fractions are distributed relative to one another depending on the crude oil's source, age, geological history, migration, and other factors [13]. Despite the complexity, petroleum compounds can be broadly classified into two main categories: hydrocarbons (measured as THP) and non-hydrocarbons. Figure 1 summarises the different categories and subclasses of total petroleum hydrocarbons. Crude oil has been separated in several ways, yet it is often a complex mixture of molecules such as B. Jet fuels contain more than 300 different hydrocarbon compounds. Its spills are generally the most severe type of pollutants that pollute the environment, such as soil, water, and ocean. In this overview, much emphasis is placed on soil and water. Leaking and routine washing have been identified as common sources of contamination. Others result from underground storage tanks, offshore platforms, wells, shipping accidents, ruptured pipelines, and natural oil spills.

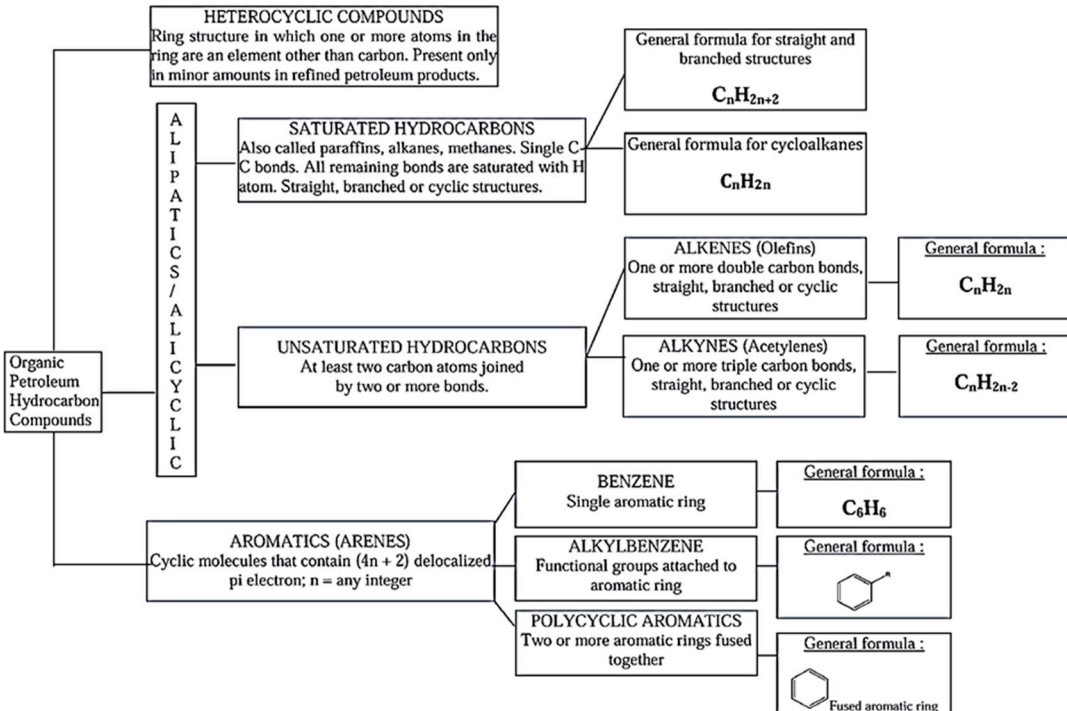

**Figure 1.** Summary of the different categories and subclasses of total petroleum hydrocarbons. Adapted and modified from Hidayat and Turjaman [11].

## 2.1. Types of Hydrocarbons Degradation

For several decades, enzymatic reactions involved in the aerobic and anaerobic degradation of hydrocarbons through microbes have been considerably reviewed [14]. This section highlights the aliphatic, aromatic, resin, and asphaltene degradation.

### 2.1.1. Aliphatic Degradation

Aliphatics, particularly alkanes, constitute a significant group in crude oil, and their exclusion from oil-contaminated fields has become an ecological concern and is considered beneficial for improving recovery [11]. Microbial degradation of alkanes could occur in several ways. Among them are terminal oxidation, subterminal oxidation, and via alkyl hydroperoxide, the enzymatic and non-enzymatic reactions taking place in these processes.

In general, aliphatic degradation has been classified in the following degree of increasing susceptibility: cyclic alkanes < branched alkanes < n-alkanes.

### 2.1.2. Aromatic Degradation

In this type, there are one or more aromatic rings; benzene is the simplest among them. Aromatics present in crude oil, with or without alkyl substituents, and their fractions are considered the second major group after the aliphatic fraction in crude oil. Aromatics with two or more fused benzene rings are classified as polycyclic aromatic hydrocarbons (PAHs), comprising a large group of xenobiotic pollutants that are common, persistent, and recalcitrant pollutants. They are reported to be potentially dangerous as some of them are highly mutagenic or carcinogenic [15].

### 2.1.3. Resin and Asphaltene Degradation

Both the resin and asphaltene fractions contain polar non-hydrocarbon chemicals in contrast to the aliphatic and aromatic fractions. The former is mostly composed of carbon and hydrogen, with traces of nitrogen, sulfur, and oxygen. Asphaltenes are high molecular weight chemicals that are insoluble in solvents like n-heptane. However, resins are polar molecules that dissolve in n-heptane, and asphaltenes are extremely complicated structures that are difficult to understand and improve through biodegradation [16].

### *2.2. Biodegradable Pollutants*

In the recent past, extremely toxic organic compounds were created and released either actively or passively into the surrounding [6]. These substances include insecticides, dyes, polychlorinated biphenyls (PCB), polycyclic aromatic hydrocarbons (PAH), and fuels. Synthetic chemicals such as radionuclides and metals are highly defiant to biodegradation by the instinctive flora paralleled to naturally occurring organic compounds, which are easily degraded when introduced to the surrounding.

### 2.2.1. Hydrocarbons

These can be seen in the form of aromatic or aliphatic hydrocarbons as linear linked, branched, or cyclic molecules. In its structure, the earlier isolation contained benzene ($C_6H_6$), while the aliphatic was observed in three forms: alkanes, alkenes, and alkynes [5].

### 2.2.2. Polycyclic Aromatic Hydrocarbons (PAHs)

These major pollutant classes of hydrophobic organic pollutants (HOCs) are broadly distributed in air, soil, and sediments, with industrial production as the primary source of PAH pollution. PAHs can be attributed to soils and sediments rich in organic matter, causing them to accumulate in fish and other aquatic organisms. They are likely to be transmitted to people through seafood intake. PAHs biodegradation can be viewed as both a typical pathway of the carbon cycle and a discharge of artificial contaminants from the ecosystem. However, using microbes for the treatment of PAH-polluted areas appears to be a compelling concept for the remediation of contaminated locations [8].

### 2.2.3. Polychlorinated Biphenyls (PCBs)

These are chemically synthesized chemical mixtures. Since they are non-flammable, chemically stable, have a high boiling point, and are electrically insulating. PCBs have been used for countless industrial purposes. These include electrical, heat transfer, and hydraulic hardware; solvents in paint, plastic, and rubber products; pigments, dyes, and carbonless copy paper. Environmental contamination brought on by PCBs is a growing issue because these toxic substances have the potential to be endocrine disruptors and carcinogens [6].

### 2.2.4. Pesticides

These substances or their mixture forms are intended to avoid, kill, repel, or reduce pests. Non-persistent substances degrade quickly, whereas persistent substances prevent

degradation. The most usual form of degradation occurs in soil by microbes that feed on pesticides, specifically fungi and bacteria [16].

### 2.2.5. Dyes

Dyes are expansively utilized in fabric, rubber products, paper, printing, photography, pharmaceuticals, cosmetics, and many other businesses. Azo dyes, the most critical and significant man-made dyes for commercial use, are inherently poorly biodegradable due to their structures. Anyway, dye-containing wastewater treatment techniques typically involve physical and chemical processes, such as adsorption, coagulation-flocculation, oxidation, filtration, and electrochemical processes, affecting the above processes [6].

### 2.2.6. Radionuclides

These atoms have unstable nuclei, characterized by the extra energy available to either be transferred to a newly created radiant particle within the nucleus or through internal conversion. Radionuclides are supposed to decay radioactively, leading to the emission of gamma rays and subatomic particles such as alpha or beta particles [17].

### 2.2.7. Heavy Metals

Metals must either be changed into a stable form or removed because, unlike organic contaminants, they cannot be biodegraded [6]. The remediation of heavy metals is accomplished via biotransformation are recorded to be the processes by which microbes operate, such as;

- Biosorption is the sorption of metal to the surface of a cell by physicochemical means.
- Bioleaching involves the mobilization of heavy metal by the elimination of organic acids or methylation reactions.
- Biomineralization (heavy metal immobilization by the formation of insoluble sulfides or polymer complexes),
- Intraceal reactions which is the process of forming insoluble sulfides or polymer complexes to immobilize heavy metals.
- Figure 2 shows the microbial mechanisms that affect metal bioremediation.

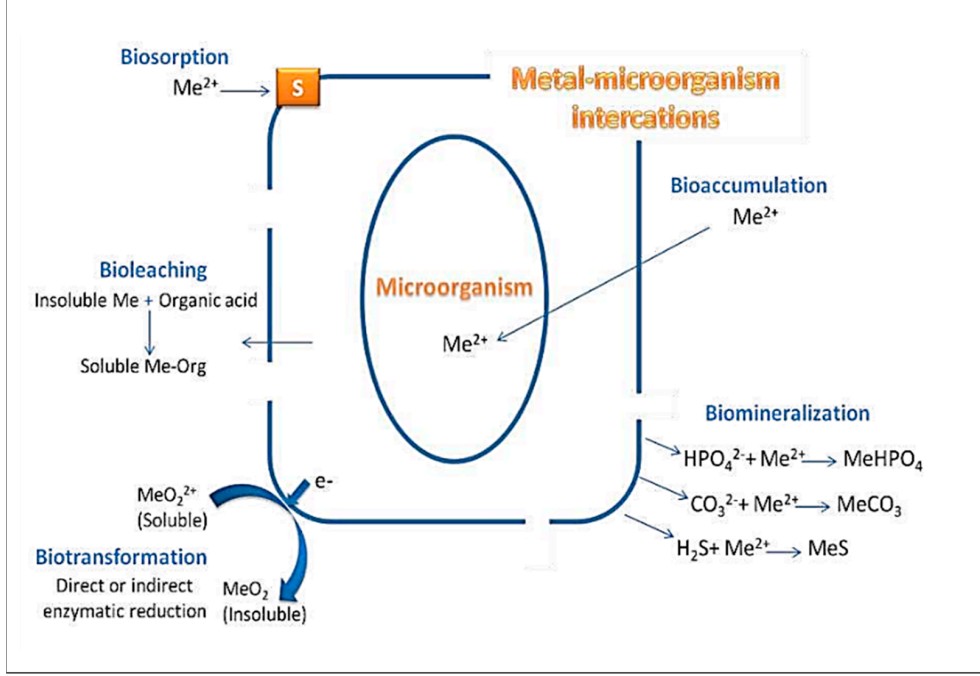

**Figure 2.** Microbial mechanisms used in metal bioremediation systems adapted and modified from Joutey et al. [6].

### 2.3. Mechanism and Influential Factors for Microbial Biodegradation

The bioremediation process required to quicken a natural biodegradation route in an economical and environmentally benign manner is timewasting [18]. As critical players in remediation, microbes tend to break down several organic contaminants due to their metabolic mechanism and adaptation to harsh environments. Nonetheless, their efficiency depends on several dynamics resulting from the chemical composition and quantity of the contaminants, their accessibility for microorganisms, and the physicochemical properties of the environs [6]. Furthermore, many of the crude oil products with higher solubility tend to acquire complex cytotoxicity towards biodegradable bacteria, though other compounds produce no substantial inhibitory properties on microbial development [18]. The factors that affect the degree of contaminant degradation through microbes are associated with the microbes and their nutritional needs (biological elements) or their ecosystem (environmental agents), which are discussed below. Figure 3 shows the pathway for the biochemical and microbial genetic pathways.

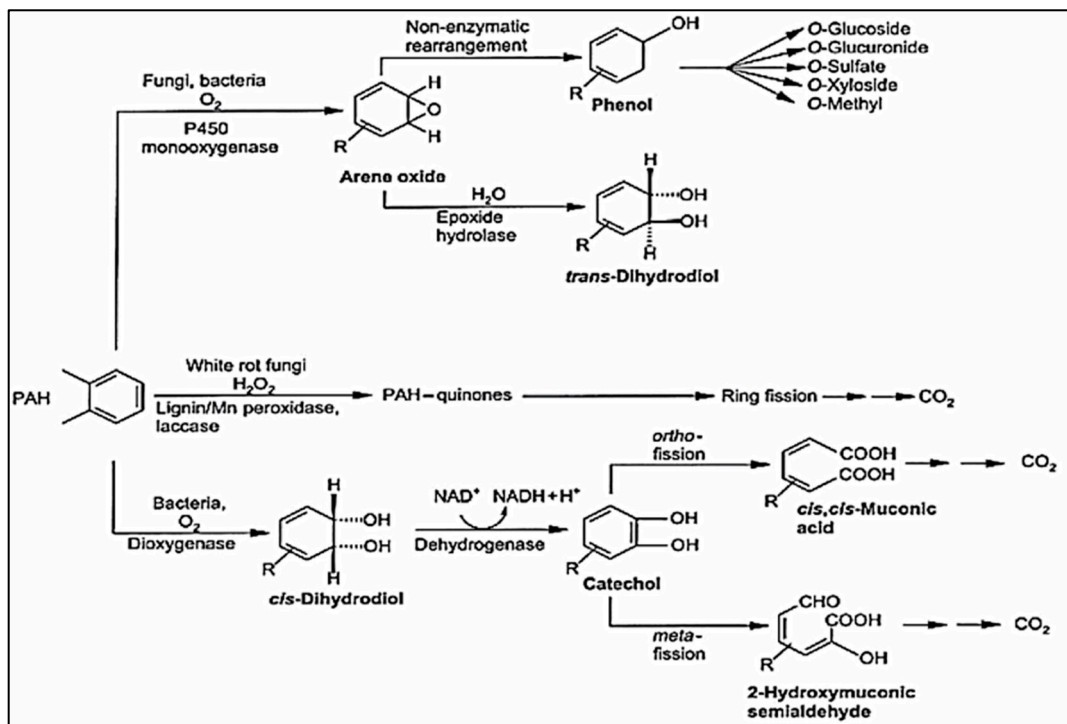

**Figure 3.** Pathway for the biochemical and microbial genetic pathway. Adapted from Hidayat and Turjaman [11].

### 2.3.1. Biological Factors

Biotic factors, which are the sole metabolic ability of microorganisms affecting the biodegradation of organic composites, include direct inhibition of enzyme activity and the multiplication process of degrading microorganisms [13]. Inhibitory activity can develop as a result of competition amongst bacteria for limited carbon sources, hostile relationships amid bacteria, or protozoa and bacteriophage preying on microbes [14]. However, the frequency of pollutant breakdown is often reliant on the amount of the pollutant and the volume of catalyst used, affecting the number of organisms that could digest the toxin and the number of enzymes created by the individual cell. For unlimited microbial development to occur, sufficient oxygen and nutrients have to be present in a usable state and in the proper proportions. Of all the influencing factors, temperature plays a dynamic role in bioremediation and thereby influences the processes.

In some cases, this indirectly affects the efficiency of biodegradation by affecting bacterial development and metabolism, fluctuations in the soil matrix, and the manner of

contaminants [18]. Crude oil, along with its byproducts, could fill in the gaps in the soil, reducing the quantity of oxygen in it. The intensity and make-up of pollutants, temperature, soil pH, oxygen content, and salinity are also greatly influenced by the bioremediation of oil-polluted lands. In this case, stunted growth of plants and microbes occurs in petroleum-rich soil, making the bioremediation process ineffective or low efficiency. With little or no oxygen availability, the degradation of aerobic microorganisms was somewhat disrupted, leading to a decrease in the bioavailability and degradation efficiency of pollutants [5]. With the presence of enzymes remaining as crucial components of the breakdown of petroleum hydrocarbons, the fluctuations in pH could also affect the enzymatic activities to decrease the efficiency of the biodegradation procedure [8].

Furthermore, high salinity and changing pH values can restrain microbial development and metabolism. Nonetheless, the deficiency of techniques to inspect the persistence and activity of organisms in the soil also limits the use of bioremediation. It has been reported that for every 10 °C drop in temperature, the biodegradation rate is reduced by about half with the same process, which is within a broad range of the optimal pH for biodegradation from 6.5 to 8 in most aquatic and marine environments, terrestrial systems [16].

### 2.3.2. Environmental Factors

Environmental factors play a crucial role in the remediation of contaminated sites. These factors, along with the microbial and physicochemical characteristics of the pollutant, determine the success of any bioremediation process. Factors such as the type of soil and the organic content of the soil influence the adsorption potential of the pollutants to the adsorbent [19]. In the sorption processes, absorption is termed as a similar process whereby a contaminant enters the substantial mass of the soil matrix. However, for both adsorption and absorption, there is a reduction in the amount of the pollutant. In this case, most microorganisms and the rate at which the chemical is metabolized is proportionately reduced [6]. Differences in penetrability of the unsaturated and saturated zones of the aquifer matrix may then influence the trajectory pattern of the fluids and contaminant passage in groundwater. As a result, the matrix's potential to distribute gases such as oxygen, methane, and carbon dioxide is reduced in fine-grained sediments, causing soils to become more soaked in water. This could have an impact on the speed and form of biodegradation that occurs.

### 3. Microbial Diversity and Ecology

Microbial ecology is critical to research in microbiology, which helps ascertain the organisms present in a specific habitat. Moreover, it helps to detect which kinds of microbes may be present within the particular site where samples were collected during the sequencing of DNAs of the microorganisms. Recently, researchers have adopted DNA-based technologies in analyzing microbial communities in organic compounds or petroleum hydrocarbon (PH) contaminated sediments, water, and soils [20]. This field of science is known as "community genomics", "economics" or "environmental genomics" [21]. It is, therefore, a study of genetic materials which are directly recuperated from samples of the environment. In its early years of study, new environmental gene sequencing cloned specific genes (16S or 18S rRNA genes), and cultivated clonal cultures were utilized to manufacture a diversified profile in a biological sample. A broad spectrum of microbial diversity used cultivation-based methods [22]. Nevertheless, recent studies have also shown that environmental samples could contain a smaller number of cultivable microorganisms than non-cultivable ones [21]. Thus, studies that focus on either polymerase chain reaction (PCR), or "shotgun" directed sequencing tend to have primarily objective sampled genes from all the sampled communities included in the study [23]. Because metagenomics can reveal the hitherto hidden diversity of microbial life, it allows microbial ecology to be scrutinized comprehensively and at a grander scale than before as DNA sequencing prices continue to decrease [21]. In analyzing the community structure of the microbes in PH-contaminated

water, sediment/soil, the total chromosomal DNA is isolated using one of numerous accessible viable DNA kits. The isolated DNA is then kept at a temperature of −20 °C for future usage. Thereafter, in a bid to analyze the microbial community structure, 16S rRNA genes are PCR amplified from the chunk DNA by utilizing PCR reaction mixture [21]. The mixture usually contains template DNA with a volume of PCR mixture between 20–50 μL, polymerase enzyme (pfu or Taq polymerases), polymerase enzyme buffer, and universal primers (e.g., 1492R/27F) each of four dNTPs [24]. Similarly, by PCR using the forward and reverse primers of 27F (5-AGA GTT TGA TCC TGG CTC AG-3) and 1492R (5-CGG CTA CCT TGT TAC GAC TT-3), respectively, the 16S rRNA region is amplified [25]. Principally, the amplification of DNA is performed under the prescribed cycling conditions. First, it starts with 1 cycle of 2 min at 94 °C. After that, it moves to 25 cycles of 30 s at 94 °C, followed by 30 s at 55 °C, then 1 min at 72 °C, and finally with the last cycle of 10 min at 72 °C [21]. After DNA amplification, PCR products, also known as 'amplicons', are tested to ascertain the particular length of the DNA PCR or DNA amplicons by a 2% agarose gel [26]. They are then purified to sequence cloned specific genes. Combo kits, which are used for the extraction and purification of gel, are commercially ubiquitous. In a single step, the combo kits can perform both purification and gel extraction of amplicons. ITS (internal transcribed spacer) regions of their complete DNA are augmented by using ITS1 and ITS2 primers in the event of analyzing the fungal community structure [27]. The purified PCR products are then utilized in the sequencing process. Myriad 'next-generation' now high-throughput methods come in handy in sequencing the genome. They include nanopore DNA sequencing, single-molecule real-time (SMRT) sequencing, Heliscope single-molecule sequencing, DNA nano ball sequencing, Ion Torrent semiconductor sequencing, solid sequencing, Illumina (Solexa) sequencing, 454-Pyrosequencing, polony sequencing and massively parallel signature sequencing (MPSS) [28,29]. The comparison of the selective high-throughput sequencing methods is shown in Table 1.

**Table 1.** Comparison of selective high-throughput sequencing methods. Adapted from Raju & Scalvenzi [21].

| Technique | Read Length | Precision | Running Time | Cost Per 1 Million Bases Pairs in US$ | Merits | Demerits |
|---|---|---|---|---|---|---|
| Synthesis sequencing | 50–500 bp: HiSeq 2500 | 99.9% | 1–11 days | 0.005–0.15 | High sequence yield potential | High concentrations of DNA require high-cost equipment |
| Ion semiconductor (Ion Torrent) | 400 bp | 98% | 2 h | 1 | Fast, high-cost equipment | Homopolymer errors |
| Pyrosequencing | 700 bp | 99.9% | 24 h | 10 | Fast, long read size | Homopolymer errors, runs are costly |
| Chain termination (Sanger) | 400–900 bp | 99.9% | 20 min to 3 h | 2400 | Useful for many applications, long individual reads | Time-consuming step of PCR, Costly |

Sequencing is complete, as seen in Figure 4; they are then analyzed using bioinformatic methods. Gas chromatography-flame ionization detection (GC-FID) is the most frequently used procedure to detect residual PHs during microbial degradation. In gas chromatography, nitrogen, hydrogen, or helium are inert gas carriers [21]. Carrier gas keeps the aqueous liquids or gas mixtures measured in a capillary column on a detector with boiling points below 400 C. In the case of complex mixtures, this enables better resolution of the components. This method monitors the substance of TPHs in the C10 to C40 range (n-alkanes) from solids such as waste and soil. The GC-FID with detection limits of 10 mg TPHs per kg soil is used

for both qualitative and quantitative applications. Another technique that mixes GC with mass spectroscopy (MS) is GC-MS [30]. Due to its flexibility in calculating PAH and TPH, the MS is considered a ubiquitous detector. Another analytical approach to characterize PHs is infrared spectroscopy (IR). The method is such that a spectrum is formed at the point where it holds energy in the IR region of the electromagnetic spectrum with bending and stretching vibrations associated with a molecule [22,29]. Hydrocarbon derivative spectra mainly derive from overtones or combinations of C-H stretching vibrations of the aromatic C-H or terminal CH3 and saturated CH2 functional groups [21,30]. Therefore, IR-based detection is beneficial in elucidating stem PHs and remaining functional groups during microbial degradation. In the biopile system of contaminated crude oil desert soils, Fourier transforms infrared spectroscopy (FTIR) and gravimetric methods have also been used to measure TPHs [31].

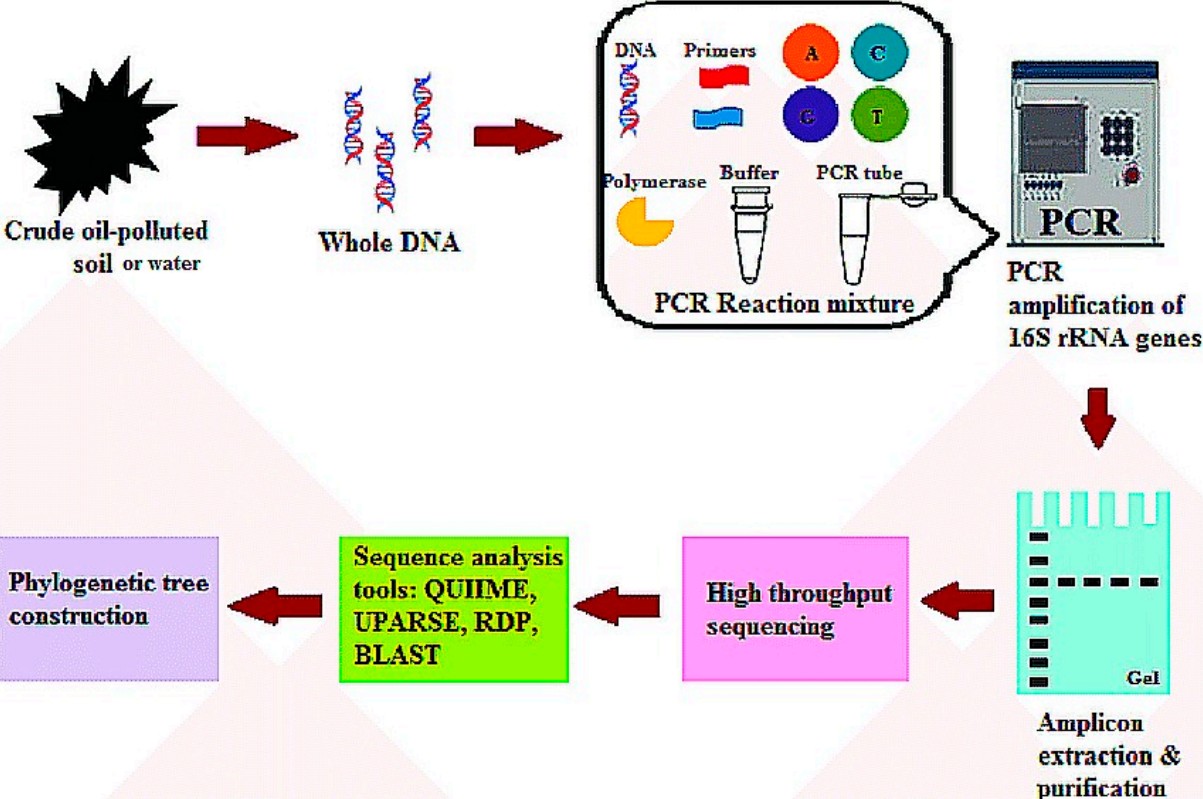

**Figure 4.** Schematic representation of microbial community analysis of crude oil-polluted soil using high throughput sequencing methods. Source: Raju & Scalvenzi [21].

## 3.1. Microbial Degradation Mechanism

The most complete and fastest degradation of organic pollutants occurs under aerobic conditions [21]. The primary intracellular attack of pollutants is activation, and an oxidative process, as well as the integration of oxygen, is the enzymatic instrumental reaction catalyzed by peroxidases and oxygenases [30]. Organic pollutants gradually convert peripheral degradation pathways into intermediate stages of the central intermediate metabolism, e.g., the tricarboxylic acid cycle [21]. The biosynthesis of biomass cells occurs on the central metabolites, e.g., pyruvate, succinate, and acetyl-CoA. The saccharides needed for diverse growth and biosynthesis are synthesized via gluconeogenesis. Specific enzyme systems could also be used to mediate the degradation of PHs. Other microbial degradation techniques utilized are the development of biosurfactants and the microbial attachment of cells to the substrates [31]. PHS can be selectively metabolized by a microbial consortium of strains or a single strain of microorganisms belonging to different or the same genera [32]. The consortium showed a way to degrade or metabolize PHs more than the

individual cultures [9,33]. Table 2 shows some microorganisms used in the biodegradation of petroleum products.

**Table 2.** Microorganisms are used in the biodegradation of petroleum products.

| Fungi | Yeast | Algae | Bacteria |
|---|---|---|---|
| Verticillium | Yarrowia | Oscillatoria | Streptomyces |
| Varicospora | Trichosporon | Agmenellum spp | Sphingomonas paucimobilis |
| Penicillium | Torulopsis | Selenastrum capricornutum | Rhodococcus spp. |
| Luhworthia | Sporobolomyces | Pseudomonas migulae | Pseudomonas |
| Gliocladium | Saccharomyces | Sphingomonas yanoikuyae | Nocardia |
| Fusarium | Rhodotorula | Chlorella sorokiniana | Mycobscterium spp. |
| Dendryphiella | Pichia | Chlorella vulgaris | Flavobacterium |
| Cunninghamella | Hamsenula | Scenedesmus platydiscus | Corynebacterium |
| Corollasporium | Debaryomyces | Scenedesmus quadricauda | Burkholderia |
| Cladosporium | Cryptococcus | Selenastrum capricornutum | Brevibacterium |
| Aspergillus | Candida | Prototheca zopfii | Bacillus |
| Phanerochaete chrysosporium | | Nitzschia sp. | Alcaligenes |
| Bjerkandera adusta | | | Alcanivorax |
| Pleurotus ostreatus | | | Acinetobacter |
| Trichocladium canadense | | | Achromobacter |
| Fusarium oxysporum | | | Pseudomonas aeruginosa |
| Achremonium sp. | | | Pseudomons fluoresens |
| | | | Haemophilus spp. |
| | | | Paenibacillus spp. |
| | | | Agrobacterium |

### 3.2. Microbial Influential Factors

Microbial degradation is environmentally friendly and inexpensive compared to other forms of petroleum hydrocarbon degradation. Thanks to the adaptability of microorganisms to hostile locations and their metabolic machinery, they can break down numerous organic contaminants in soil and waterbodies. They, therefore, perform an important function in site remedy. Nonetheless, some factors could limit their ability and ability to mine petroleum products fully. El Fantroussi and Agathos [34] state that the features that can affect the efficiency of microbial degradation include the physicochemical properties of the environment, the availability of microorganisms, and the concentration and chemical attributes of the contaminants. Therefore, they could be influenced by the environment (environmental factors) and the microorganisms and their nutritional needs (biological factors) [6]. Al-Hawash et al. [35] add that nutrients, pH, oxygen, and temperature are some of the factors that can influence microbial activities. The metabolic ability of a microorganism depends on biotic factors. These biotic factors include the proliferation process of degrading microbes and the express inhibition of enzymatic actions. The obstruction thus occurs when microorganisms are predated by bacteriophages and protozoa, antagonistic interactions between the microorganisms, or competition among microorganisms for inadequate carbon sources [6]. The amount of catalyst present and the concentration of the contaminant affect the degradation rate of the contaminant. In this case, the number of catalysts is the enzyme produced by each cell and the number of organisms capable of metabolizing the pollutants.

Furthermore, the degree of processing of pollutants is, to a large extent, a component of the enzymes involved and their affinity for the pollutant, and the accessibility of the pollutant [30]. Another factor that can affect the microbial degradation of petroleum products is the presence of microbes that can degrade the pollutants, including algae, fungi, yeast, and bacteria [35]. These microorganisms rely on oil spills as a food source, especially in places where there are gas stations, oil fields, ports, shipping lanes, and crude oil wells. In addition, the availability of sufficient amounts of oxygen and nutrients in usable forms and proportions ensures that unrestricted growth of microbial activities takes place. Oxy-

gen concentration was found to be a rate-limiting variable for ambient PH degradation. The availability of oxygen depends on the available usable substrates, whether the soil is water-saturated or not, soil type, and microbial oxygen consumption rates that can reduce oxygen [36,37]. Some research has also shown microorganisms' anaerobic decomposition of PHscan occurs at nominal rates [36]. Recent research has also confirmed that in the absence of molecular oxygen, soil and sludge microbial consortia can metabolize alkyl-unsubstituted and substituted aromatics to xylene, toluene, naphthalene, acenaphthene, 1,3-dimethylbenzene and benzene [35]. In addition, studies have also reported that anaerobic biodegradation of PHs is less and slower than aerobic biodegradation. Therefore, substrate oxidation through oxygenation in the microbial catabolism of all aromatics, cyclic, and aliphatic compounds is a key phase in bioremediation [37].

### 3.2.1. Nutrients

This is an essential element for successfully degrading contaminants, including iron, nitrogen, and phosphorous in some cases. Nonetheless, they can also become an inhibiting factor in the biodegradation process. In freshwater and marine environments, oil spills decrease phosphorous and nitrogen levels and a dramatic increase in carbon levels, which impacts the biodegradation process [38]. Moreover, if phosphorous and nitrogen levels are low, the wetlands cannot provide nutrients due to the intense demands of nutrients by the plants. Consequently, additional nutrients are necessary to aggrandize contaminants' biodegradation [39].

### 3.2.2. Concentration

The over-concentration of excess nutrients can curtail the biodegradation process. Zafra et al. [40] found that the concentration of the pollutant had a particular pressure on petroleum-degrading organisms. The high PAH levels were restricting the development of microorganisms that built up a response against PAHs concerning the alterations of mycelia and sporulation pigmentation and the structure of the cell membrane. Balaji et al. [41] conducted on the various sources of carbon to produce lipase by Mucor racemosus, Lasiodiplodia theobromae, and Penicillium chrysogenum showed that cellulose and sucrose induced the highest activity in the mentioned species. Additionally, the study discovered that yeast extract was a superior key contributor to high lipase levels. Moreover, the pyrene-degrading efficiency and the strain growth were improved in comparison to the control experiment after 7 and 14 days of incubation by Hypocrea/Trichoderma using pyrene as a carbon source when 0.1% sucrose or lactose or 0.02% yeast extract was studied during PAHs degradation [40].

### 3.2.3. Temperature

This can as well influence the biodegradation process by controlling the rates of catalyzed reactions of the enzymes. In this instance, it affects the chemical and physical compositions of the PHs. When temperatures are low, the degradation rate tends to decrease due to the reduced rates of enzymatic activities [42]. However, high temperatures between 30 °C to 40 °C ensure that the rate of hydrocarbon metabolism reaches its peak. The highest rates of degradation occurred at 30 °C to 40 °C for soil, 20 °C to 30 °C for marine and 15 °C to 20 °C for freshwater environments.

### 3.2.4. Bioavailability

This has been found to affect the microbiological, chemical, and physical factors affecting the rate and extent of biodegradation. It refers to that part of a compound in soil that can be taken up or modified by living organisms. The constraints can fundamentally affect pH, the microbial network, and the degree of hydrocarbon degradation in hydrocarbon bioavailability. PHs are categorized as hydrophobic organic contaminants with restricted bioavailability. These chemicals have low water solubility, making them impervious to biological, chemical, and photolytic degradation [43].

### 3.2.5. pH

The pH of the environment can also influence biological processes, e.g., Enzyme activities, catalytic reaction equilibrium, and cell membrane transport. Its variability must be taken into account, enhancing biological treatment approaches. Unlike aquatic ecosystems, most heterotrophic bacteria grow in a neutral to alkaline pH, and soil acidity can also vary from pH 2.5 to pH 11 in alkaline deserts. However, heterotrophic bacteria and fungi generally favor a near-neutral pH, although fungi can tolerate acidic conditions. Studies have also shown that pH 6.5 leads to microbial mineralization of octadecane and naphthalene [35]. The studies again showed that while the mineralization rate of naphthalene was unchanged when the pH was increased from 6.5 to 8.0, octadecane mineralization increased remarkably. Pseudomonas aeruginosa biodegrades crude oil to the highest degree in the water and mud samples, with pH values of 8.0 and 7.8, respectively [44]. Moreover, Pawar [45] found that the most suitable pH for the degradation of all soil pH is 7.5, and the degradation of phenanthrene in liquid media at pH 6.5–7.0 was beneficial [46].

Biological treatment is usually about removing pollutants and toxins from the confined environment with microorganisms. Bioreactors have been used to treat contaminated PAH and remove oxy-PAHs such as PAH ketones, quinones, and coumarins [47]. These compounds can be configured by chemical oxidation and phototransformation of PAHs and are formed via the metabolism of microorganisms to PAHs. The availability of water for microbial growth and metabolic processes can affect the biodegradation of hydrocarbons in terrestrial ecosystems. Biodegradation reaches its optimal level with 30 to 90% water saturation in the oil sludge [35]. Tar globule deposits could also limit microbial degradation to hydrocarbons. In addition, a link exists between the rate of mineralization and the salinity of PAHs in estuarine sediments. Studies reported an overall decrease in microbial metabolic rates when salinity increased to 3.3–28.4%, leading to the evaporation of salt ponds [35,37]. Therefore, salinity affects biodegradation as it affects microbial diversity and growth [48]. Moreover, it has an unfavorable effect on some necessary enzymes involved in the breakdown of hydrocarbons [49]. Table 3 shows the microbial factors influencing the biological degradation of mineral oil products.

**Table 3.** Influential factors of microbial ecology adapted and modified from Okoh [50] and Singh & Chandra [51].

| Parameter | Condition |
|---|---|
| Pressure | Monitoring was done at 10 atm and 495 or 500 atm for the breakdown of hydrocarbon feedstock by a culture broth of deep sea sediment bacteria. |
| Salinity | The hydrocarbon breakdown rate decreased with increasing salinity between 3.3% and 28.4%, and the findings were linked to a gradual decline in microbial metabolism activities. |
| Bioavailability | Due to their capacity to boost bioavailability, research into bio-surfactants and microorganisms that produce them has been done on a number of occasions. Surfactant application in oil-contaminated locations may have a neutral, inhibiting, or stimulatory effect on bacteria decomposing oil molecules. The degree of alkane degradation is impacted by the introduction of foreign oil spill dispersants or non-surfactants. |
| Soil moisture, alkalinity, and acidity (pH) | In an aquatic setting, the water potential can be as low as −0.98 and as high as 0.0–0.99 in soil. The best rate for oil sludge biodegradation in soil is between 30% and 90% of saturated water. Extreme pH range in soil: mine soil −2.5, alkaline desert −11.0. Most heterotrophic fungi and bacteria favour a pH neutral. |

**Table 3.** *Cont.*

| Parameter | Condition |
|---|---|
| Nutrients | Additional nutrients enhance the biodegradation of oil pollution. Numerous articles have been published on the detrimental effects of excessive NPK use in the degradation process of hydrocarbons. It has been discovered that organic fertilisers, such as chicken manure, speed up the biodegradation of contaminated soil. |
| Oxygen | Photo-oxidation increased the degradation rate of oil hydrocarbons by improving their bioavailability and, as a result, their microbial actions. The presence of usable substrates, the soil type and the rate of $O_2$ utilisation by microbes that can deplete oxygen all influence oxygen levels. During the biodegradation of crude oil in soil, oxygen concentration is thought to be a rate limiting parameter. In the early stages of aromatic, cyclic, and aliphatic hydrocarbon catabolism by fungi and bacteria, oxygenases oxidise the compound, which requires $O_2$. |
| Temperature | Excellent hydrocarbon bioremediation has been observed in temperate regions with psychrophilic surroundings. The maximum rate of degradation occurs at temperatures ranging from 15 °C to 20 °C in marine, 20 °C to 30 °C in freshwater, and 30 °C to 40 °C in soil. Enzymatic activity declines at lower temperatures, which also slows the rate of biodegradation. Low temperatures increase oil viscosity and decrease the volatility of harmful low molecular weight hydrocarbons, reducing microbial degradation. |
| Structure and composition of hydrocarbon | The vulnerability of hydrocarbons to microbial attack is proportional to the degree of degradability. Saturates-light aromatics-high molecular weight aromatic polar compounds or n-alkanes-branched alkanes-cyclic alkanes |
| Weathering | Photo oxidation, evaporation |
| Microorganisms | PHC degraders could be scarce or unavailable |

## 4. Emerging Biodegradation Technologies

Petroleum hydrocarbons have adverse effects on plants and animals. They contaminate the environment and challenge academic and industrial experts to find mutually agreed remediation options to avert the impact on petroleum-contaminated media. It is worth noting that about 60–90% of the chemical composition of petroleum is biodegradable, which is why research efforts are directed toward new degradation technologies, in particular, the bioremediation of petroleum hydrocarbons in soil and water [35]. Some of these remediation technologies are shown in Figure 5. The subsequent rapid and complete degradation occurs under aerobic conditions for most organic pollutants [52]. The first intracellular organic pollutant attack occurs due to oxidation and activation, and oxygen implementation is the critical factor in enzymatic motivation by peroxides and oxygenates [53]. Peripheral degradation pathways convert organic pollutants into intermediate stages of the central intermediate metabolism, such as the tricarboxylic acid cycle. Cell biomass metabolism is produced by metabolites of major precursors such as acetyl-CoA, pyruvate, and succinate [46]. The saccharides required for various biosynthesis and growth are synthesized by gluconeogenesis. The pH reduction may be possible through a specific enzyme system. Other processes are also involved, such as microbial cell adhesion to substrates and the development of biosurfactants [54]. pH values can be preferentially metabolized by a single strain of microorganism or by a microbial strain synthesis relevant to the same or different genera. The consortium has shown that there are more ways than individual cultures to metabolize or degrade PHs.

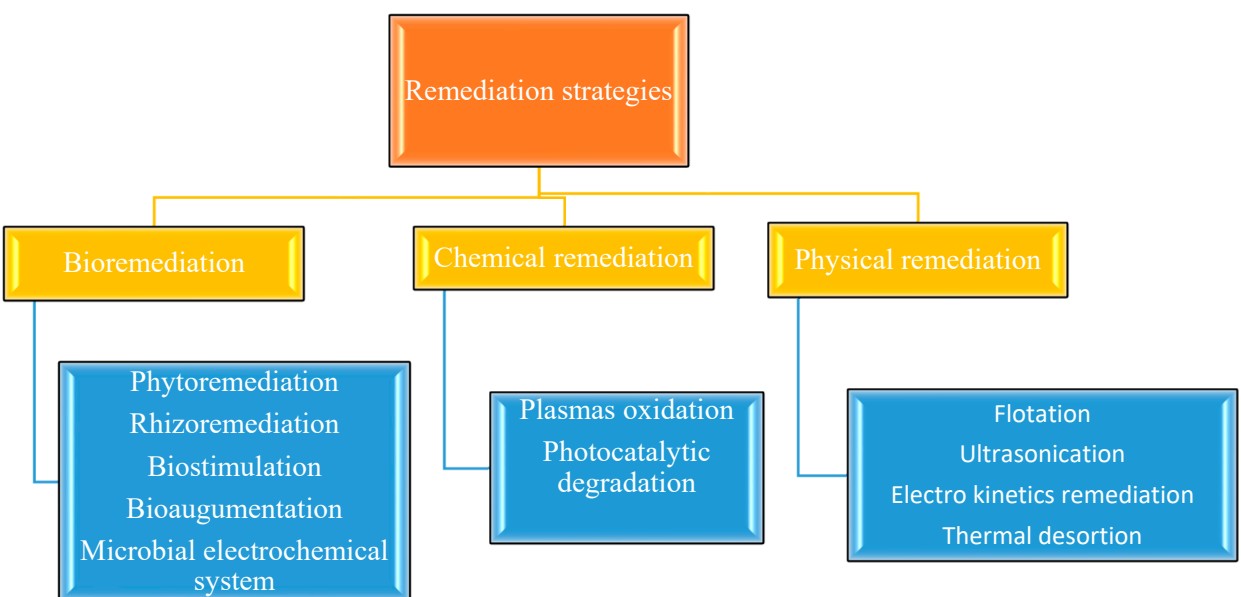

**Figure 5.** Diagram of soil remediation techniques for oil-contaminated areas.

### 4.1. Physical Remediation

First, the physical remediation approach requires recognizing the physical properties of the contaminant or the medium to introduce countermeasures to remove, isolate, or contain the contamination [55]. These countermeasures include the extraction of vapor (evaporation), flotation, ultrasound, remediation of electro kinetics, thermal desorption, and biochar adsorption [56]. An inducing volatilization mechanism is used in soil vapor extraction on the nonaqueous-phase liquid to transport volatile organic compounds for further treatment from a low-concentration subsurface to the surface of the soil [57]. The process could use an extraction well to create a concentration gradient that will induce the flow of volatile or semi-volatile pollutants to a higher concentrated point for removal from the soil [58]. The extraction of soil vapor allows the follow-up of oxygen in the soil particles, thereby allowing microorganisms to thrive in the environment. It should be noted that soil vapor extraction performance is directly influenced by soil properties such as porosity, density, texture, operating conditions, and properties of pollutants [59]. In situ soil venting and soil vacuum extraction are other known terminology used for the extraction of soil vapor.

#### 4.1.1. Flotation

The contaminated soil or water surface properties are used in this technology to extract the oil from the soil or water using a gas-liquid-solid device. The flotation mechanism depends on (i) collision between pollutants and bubbles, (ii) bubble-pollutant shape with pollutant and bubble attachment, (iii) bubble-pollutant flotation depending on buoyancy difference and bubble-pollutant-pollutant detachment [56]. Flotation is defined as simplicity, low operating costs, and high performance in removing pollutants. At low rates of descent, it can also separate very small or light particles. Despite this, a large amount of wastewater was generated during the flotation process. Moreover, productivity has significantly declined from aged or weathered polluted soil.

#### 4.1.2. Ultrasonication

Ultrasound helps in the desorption of the pollutant and facilitates the formation of solid oxidants and hydroxyl radicals (OH), which increase the pollutant removal efficiency [53]. Without chemicals, ultrasound removes dangerous pollutants. Furthermore, heat and mass transfer processes would be improved by spot heating and vigorous agi-

tation. Nevertheless, the structure is pricey due to the increased energy use for acoustic generation.

### 4.1.3. Electrochemical System

Electrokinetic remediation uses direct electrical current between suitably spaced electrodes (cathodes and anodes) that form an electric field embedded in petroleum-contaminated soil. The fluid medium began to flow preferentially toward the cathode in the electric field as a result of the voltage potential gradient that was created, dragging the contaminant behind it in the bulk flow. The value of remediation of electro kinetics is focused on the rate of implementation and low cost of activity [53,59]. In addition, during remediation, the electro-osmotic flow is persistent over the entire soil mass, which is ideal for soils with low permeability. The electro-kinetic method, however, is futile at low concentrations of contaminants. After a long period of time, the soil pH and hot spots around the electrodes were adjusted [53,54].

### 4.1.4. Thermal Desorption

This phenomenon is based on temperature modulation to raise the vapor pressure of the pollutants in which the pollutants were volatilized and subsequent desorption from polluted soil [60]. In high heat conditions, thermal desorption effectively removes the oil contaminants. Furthermore, thermal desorption releases into the atmosphere little or no contaminating gas [61]. Nevertheless, most volatile pollutants can be removed by thermal desorption.

### *4.2. Chemical Remediation*

This allows soil pollutants to be deposited or pre-processed quickly. In chemical oxidation remediation, oxidizing agents can bring about the rapid and complete chemical degradation of petroleum pollutants [61,62]. Contaminants are chemically transformed into non-hazardous, biodegradable, or less volatile compounds that are more stable, a bit static, or chemically inert.

### 4.2.1. Plasmas Oxidation

This is a highly competitive technology for the remediation of soil contaminants. There are many issues with plasma technology used in soil remediation that uses pulsed corona and dielectric barrier discharge [63]. Plasma is an electrically neutral macroscopic mixture consisting of various ions, electrons, atoms, molecules, and neutral unionized particles. In the generation of plasma by ionization, several active constituents such as $O_3$, $H_2O_2$, hydroxyl radicals ($OH^-$), and high-energy electrons were produced in which a strongly oxidizing environment was developed for oxidative contaminant decomposition [59,63].

### 4.2.2. Photocatalytic Degradation

This is effective in decomposing soils with PAHs. This technology employs a semiconductor metal oxide as a catalyst for directly breaking down chemical molecules into small molecules [64]. A valence band with stable power electrons and an empty conduction band of higher energy are contained in the semiconductor molecule. The photocatalytic reaction, which produces holes ($h^+$) in the valence band and electrons ($e^-$) in the conduction band on the femtosecond timescale, can initiate radiation absorption. Hydroxyl radicals ($OH^-$) and superoxide radical anions ($O_2$) are generated to break down organic impurities during the photocatalytic phase. However, the properties of light absorption, moist matter content, and soil moisture content can affect photocatalytic degradation [58,64].

### *4.3. Bioremediation*

The degradation of hydrocarbon compounds into smaller organic and inorganic compounds via the action of biological agents (microorganisms, plants, or plant residues) and the restoration of soil or water to its original state is generally known as Bioremedia-

tion [60,64]. This degradation technique utilized the intrinsic ability of the native microorganism to act upon the pollutants while creating conditions that enhance the increased biodegradation rate. Biological processes are often used as a substitute for chemical or physical clean-up of oil spills since most bioremediation techniques require less equipment or labor than other processes [62,64]. These are classified based on the location. The classification is in situ (on-site where the pollution took place) and ex-situ (outside the location of the pollution) bioremediation [60]. The in-situ bioremediation technique allows microorganisms to work efficiently as the local environment benefits their growth with no phase of adaptation required. Ex situ bioremediation techniques, alternatively, are primarily based on the physical exploitation of the pollutants without microorganisms' direct involvement in the remediation measures [60,61]. Conversely, the In situ bioremediation technique is preferable as it is cost-effective compared to the cost of transporting contaminated soil out of the site, likewise, it is more effective in the remediation of a large area. This section gives an overview of recent bioremediation technologies and the operating factors aiding the success of each technology.

### 4.3.1. Agricultural Remediation

The composition of pollutants, concentration of pollutants and microorganism present, temperature, soil pH, salinity, and nutrient availability are the various factors influencing the success of the bioremediation technique [63]. Generally, microorganisms are present in all sub-surface soil layers, but due to the composition and concentration of pollutants, the microbial biota present may not be effective in treating the contaminated sites, therefore, may need oxygen (for aerobic biodegradation), nutrients (to enhance the growth of microbes) and moisture (to aid microbial activity) [65]. The optimum soil moisture for biodegradation to be effective is within the range of 12–30% weight composition and 40–85% field capacity. pH within 6–8 is optimum for effective bioremediation, but the remediation process may accommodate pH within 4–8.

Land farming also referred to as land recovery, is a method of remediating aboveground soil that lowers the concentration of crude oil components through the degradation process [65,66]. Traditionally, land farming involves spreading polluted soils on the ground surface of a treatment site in a thin layer and enhancing aerobic microbial activity inside the soil to accelerate naturally occurring biodegradation mechanisms [66]. This phenomenon is usually used to remediate hydrocarbon-polluted sites, including polyaromatic hydrocarbons. As a result, the two remediation mechanisms involved in the removal of pollutants are biodegradation and volatilization [66]. It decreases the concentration of petroleum hydrocarbon constituents predominantly via bacterial-mediated biodegradation, while volatilization, abiotic, and fungal-mediated processes may also play a role [54]. Therefore, building a suitable land recovery scheme with a waterproof liner minimizes the seeping of pollutants into neighboring areas during the operation [46]. This approach is considered based on the depth of the pollutants below the surface. If a pollutant lies < 1 m below the ground surface, bioremediation can continue without excavation, while pollutants that lie > 1.7 m must be transported to the ground surface to effectively improve bioremediation [67]. At large, excavated contaminated topsoils are prudently applied to a fixed layer of support above the ground surface to allow utochthonous microorganisms to aerobically biodegrade the pollutant [68]. Tillage-triggering aeration, nutrient adding (nitrogen, phosphorus, and potassium), and irrigation are the key processes that stimulate autochthonous microorganism activity [57]. This is very efficient when environmental situations are suitable for microbial development and activity. Its application often involves improving certain environmental parameters, including humidity, pH, oxygen, and nutrient availability [69]. Furthermore, because of its structure and pollutant discharge process (volatilization), it is not appropriate for treating soil contaminated with harmful volatiles, particularly in hot (tropical) climatic areas. These and other constraints make land-based bioremediation time-consuming and inefficient when likened to other ex situ bioremediation approaches.

### 4.3.2. Biopile

Biopiling, also branded as bioheaps, biocells, or biomounds, is an ex-situ bioremediation technique commonly used to address a wide variety of petrochemical pollutants in soils and sediments. This entails building piles of contaminated soils or dried sediments and encouraging aerobic microbial populations to degrade the material by fostering ideal or nearly ideal growth conditions inside the pile [65,69]. This includes aeration, adjustment of pH and humidity levels, nitrogen and phosphorus addition, and introduction of heat. As a result of these optimal conditions for growth, the increase in microbes' activity disintegrates bioavailable pollutants. Biopile innovation is effective for removing difficult-to-desorb natural impurities or moderately biodegradable natural constituents. Nonetheless, the biopile system has its drawbacks, such as conserving plenty of space, viable engineering, installation and repair, non-availability of power supply in isolated zones, and reduced microbial activity resulting in heat generation [57,65]. Likewise, constituents such as volatiles are dissipated during the process, and it is limited for hydrocarbon concentrations exceeding 50,000 ppm.

### 4.3.3. Bioreactor

The conversion of polluted media via a series of biological processes in a vessel (bioreactor) to a specific product is an effective bioremediation technique. Engineered bioreactors designed to provide optimum conditions for microbial growth and biodegradation were designed for use in bioremediation strategies to improve the various anticipated objectives [61]. Bioreactors can be in batch, continuous, fed batch, and multistage mode and are aimed at optimizing microbial processes concerning polluted media as well as the kind of pollutant. Slurry bioreactors offer an ex-situ eco-sustainable way to remediate most soils and sediments from petroleum hydrocarbons and explosives when formed into a slurry [61,65]. The most fundamental biofilter bioreactor consists of a sizeable media bed where the microorganisms pass through the pollutants for degradation. Biofilters are among the earliest known bioremediation techniques used in the environment. The trickling filter is an example of a biofilter, which has a wide potential for treating various wastewater or liquid-formed waste [65,69]. Microorganisms grow on the surface of the packaging material in biofilm forms and are responsible for the degradation of effluent pollutants. Membrane bioreactors use a membrane to create a biological filtration system [65]. The membrane offers a barrier that excludes the solid from the liquid component while ensuring good effluent quality. The fouling of the membrane has been identified as an important drawback in its utilization for the bioremediation process, as well as the cost of the membrane, which makes the process expensive.

### 4.3.4. Composting

Composting is a method involving the stacking of polluted soil along with organic substances [21]. Often these are added to complement the measure of nutrients and organic matter readily degradable in the topsoil, stimulating bacterial growth by adding nutrients results in efficient biodegradation within a relatively brief timeframe [21,69]. Compositing provides a greater oleophilic microbial population and higher temperatures making it more promising than land farming which is based exclusively on native soil biota. Likewise, compost manure, which is useful for agricultural purposes, is generated as an end product [69]. The benefits of composting include the enrichment of soil quality and characteristics as well as its ecofriendly nature to the contaminated site. Its major drawbacks include intensive monitoring of the site, it time-consuming as it may take a month as shown with some recent application in Table 4, and it is labor intensive. Toxic and odorous (green house) gas may be released into the environment as well.

### 4.3.5. Bioventing

In bioventing, oxygen-rich air is added to the soil to improve the degradation rate of contaminating organic pollutants [70]. This technique uses low-pressure air and focuses

more on the deeper unsaturated soil zone. The simple bioventing setup consists of a series of connected blowers or air compressors connected to air supply wells and soil gas monitoring wells [70,71]. Bioventing is the mildest and simplest form of bioremediation since it does not interfere with the natural environment of microorganisms [65]. Bioventing is particularly preferred for hydrocarbons with very low volatility [46]. Due to the efficient bioremediation of these petrochemicals, the volatilization rate must be kept at an optimal level, which should be lower than biodegradation rates. Low volatility likewise diminishes the likelihood of degradation as air injection through the bioventing process carries contaminants into the environment [71]. The principal benefits of this technique are the cost, the short treatment time, and the ease with which it can be combined with other techniques (such as bioslurping). Its limitation is that it cannot be applied to low-permeability soils, less effective on sites contaminated with chlorine and heavy hydrocarbons [52]. Bioventing leads to a reduction in soil moisture, which affects the degradation rate. Table 4 shows some recent applications of bioventing and the results obtained.

### 4.3.6. Bioslurping

This is the adaptation and deployment of vacuum-enhanced dewatering and biostimulation technologies for the treatment of polluted hydrocarbon sites. Vacuum-enhanced pumping allows light non-aqueous phase liquids (LNAPLs) to be extracted from the water table and released from the capillary fringe in bioslurping, reducing changes in water table elevation and avoiding smear zone formation [69]. The pumping mechanism allows LNAPLs to travel upwards to the surface, where water and air are separated. The conventional bioventing method, once free products (pollutants) have been isolated, finalizes the treatment of contaminants to complete the remediation process. This technique is cost-effective as only a limited volume of groundwater, and soil vapor is pumped at a time [66]. Some of the recovered hydrocarbons using bioslurping are toluene, xylene, paraffin, naphthalene, and olefins at polluted Greek petroleum sites. However, because the technology does not directly interact with the saturated region, there were some petroleum residues in the groundwater [70].

### 4.3.7. Biosparging

Biosparging is the direct injection of compressed air (consisting mainly of oxygen) into the saturated subsoil. As a result, the generated bubbles lead to the detachment of contaminants from the groundwater. As a result, they are transported to an unsaturated zone where the pollutants are degradable in situ through in situ bioremediation [65,70]. This technology supports the activity of native microorganisms by adding air or nutrients to the heavily polluted zone. Biosparging is mainly used to promote biodegradation activity, the direct approach being biostimulation by injection of air or nutrients [2]. It can be used in contaminated groundwater as shown in Table 4. Its major benefits are ease of equipment installation, and soil excavation is not required. Its main drawback is that the airflow direction cannot be easily predicted.

### 4.3.8. Natural Attenuation

A natural process called natural attenuation reduces the concentration and quantity of pollutants at contaminated sites. It can also be referred to as intrinsic remediation, attenuation, and bioremediation [66]. It is mainly used when the source of contamination has been removed to remediate the contaminated aquifer. It is used mainly for benzene, toluene, ethylbenzene, and xylene (BTEX) and, recently, for chlorinated hydrocarbons. The success of natural damping strongly depends on the sub-surface's geology, hydrology, and microbiology [55,66]. Natural attenuation boosts the benefits of being able to be applied to all sites, as well as producing little or no damage to the sites, and it is the most cost-effective. The main drawbacks of natural attenuation are that it is a relatively time-consuming practice because it is a non-technological procedure of biodegradation.

Long-term monitoring is imperative because there must be no danger to the environment and humans [66,69]. Table 4 shows some recent applications of natural attenuation.

**Table 4.** Recent application of some of the bioremediation techniques and the results achieved.

| Bioremediation Technology | Recent Applications | Results |
| --- | --- | --- |
| Biopiling | <ul><li>Treatment of petroleum contaminated soil via biopile system for 80 days [72]</li><li>Treatment of Kuwait petroleum contaminated soil via biowashing and biopiling system for 20 days [73]</li></ul> | <ul><li>69% maximum degradation was observed for one of the tests.</li><li>Overall reduction of 86% TPH removal was achieved, biopile system was used for 4 days for 21% removal of residual TPH after biowashing.</li></ul> |
| Composting | <ul><li>Pilot field test for removal of petroleum from an old refinery site within 90–120 days [74]</li><li>Composting of petroleum contaminated soils for 98 days [75]</li><li>Remediation of 1200 m$^3$ of saline-contaminated soil in Iran [76]</li></ul> | <ul><li>98% removal efficiency of TPH was achieved using co-composting</li><li>Between 50–79% removal of TPH was observed.</li><li>99% removal efficiency achieved after 2 months</li></ul> |
| Bioventing | <ul><li>Treatment of PH contaminated sites using 80 kg reactor for silt and loamy soil for 30 days [77]</li></ul> | <ul><li>75% degradation efficiency observed for 8o kg reactor</li></ul> |
| Biosparging | <ul><li>Treatment of shallow groundwater contaminated with benzene, toluene and ethylbenzene [78]</li></ul> | <ul><li>Degradation rate of 80% was observed with increased degradation rate via biosparging</li></ul> |
| Natural attenuation | <ul><li>Bioattenuation of soil from crude oil contaminated site [79]</li><li>Treatment of crude oil contaminated soil [80]</li></ul> | <ul><li>52% degradation of PAH achieved</li><li>40% removal PAHs achieved</li></ul> |

*4.4. Factors Influencing the Degradation of Petroleum Hydrocarbons*

Climatic factors can also have a substantial impact on the pace and extent of biodegradation (Figure 6). At spill sites, variables such as oxygen and nutrient accessibility can also be exploited to improve innate biodegradation (i.e., using bioremediation). Other variables, such as salinity, are usually uncontrollable. The large extent to which biodegradation can affect a given environment causes some difficulties in accurately predicting the success of bioremediation efforts [57,64]. Another source of uncertainty is the lack of awareness of the impact of various environmental factors on the rate and degree of biodegradation. Bioremediation is viewed as a tool to speed up the process of natural biodegradation in a cost-effective and environmentally friendly manner. However, bioremediation takes time, and the concentration and composition of pollutants, temperature, soil pH, oxygen status, and salinity are strongly influenced by the bioremediation of petroleum-contaminated soils [81]. Plants and microbes cannot thrive in oil-rich soils. In this case, it was inoperable or had poor efficiency for bioremediation.

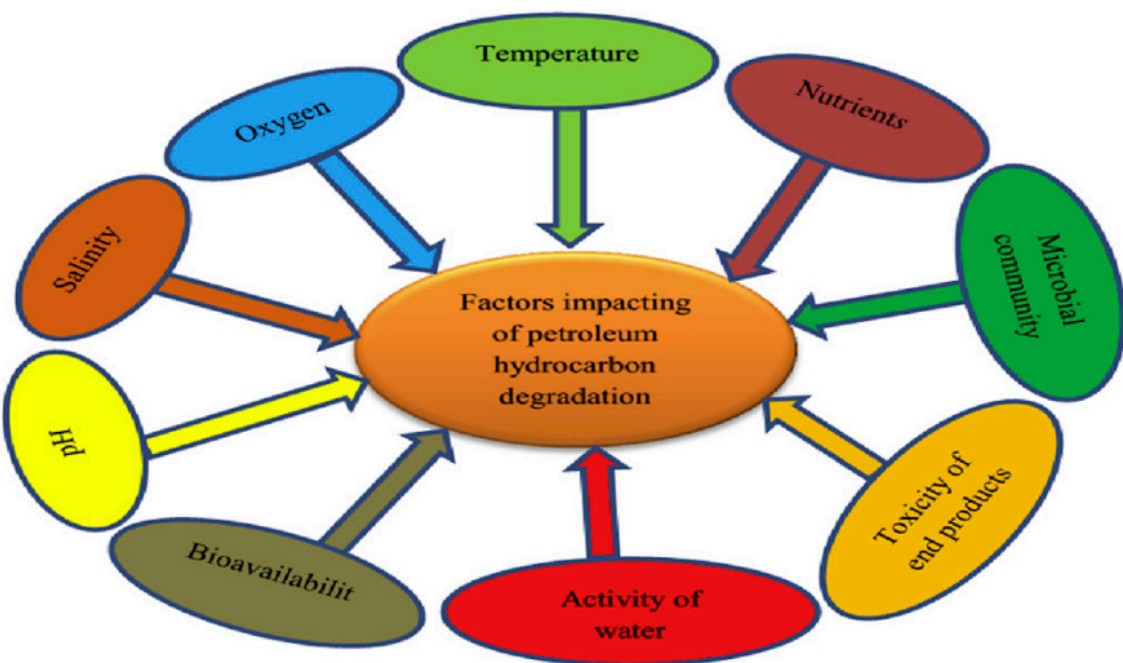

**Figure 6.** Schematic representation of factors that influence microbial remediation.

Furthermore, some highly soluble petroleum derivatives have more significant cytotoxicity for bacterial biodegradation, while other compounds do not have significant inhibitory effects on bacterial growth [60]. Temperature plays a crucial role in bioremediation and influences biodegradation reactions. Temperature can indirectly affect biodegradation performance by affecting bacterial growth and metabolism, changing soil matrix, and the mode of appearance of pollutants. Petroleum and its derivatives can fill soil voids, reducing the amount of soil oxygen [14,82]. The metabolism of aerobic microorganisms was partially disturbed under reduced or absent oxygen conditions, and the bioavailability and degradation efficiency of pollutants was reduced. Various essential enzymes are the main components of the biodegradation of petroleum hydrocarbons. To reduce the efficiency of biodegradation, changes in pH can affect the activities of the enzymes. On the other hand, changes in pH and high salinity inhibit microbial growth and metabolism. In addition, the lack of a technique to track the survival and activity of microorganisms in an acceptable environment can also limit the use of bioremediation.

*4.5. Genetically Engineered Microorganism*

Genetically engineered microorganisms (GEMs) are microorganisms modified by synthetic strategies driven by unique artificial genetic switches between the microorganisms [14,82]. Furthermore, the following protocols shall be considered throughout the GEM process (a) changing enzyme selectivity and affinity, (b) pathway evolution and rules, (c) bioprocess improvement, tracking, and manipulation, (d) bio affinity bioreporter Sensor applications for chemical detection, toxicity reduction and endpoint assessment [82,83]. Genetically engineered microbes offer the advantage of breeding microbial lines that can withstand unfriendly, frightening circumstances and be used for bioremediation in unique and complicated natural conditions [17,82,84,85]. In response, molecular approaches have been implemented to easily restrict trace bacteria such as Escherichia coli, Pseudomonas putida, and Bacillus subtilis activities [67,82,86,87]. This implies that different microorganisms should be studied for their application in enormous bioremediation. Therefore, the application of GEM is still limited to the laboratory/pilot scale; it is yet to find the application on an industrial scale because (i) its containment in terms of what may happen during post-application, (ii) it impacts the native micro-organisms [88].

Even though there has been recent progress in the technology involved in the engineering of microbes for their application in bioremediation or biodegradation, nevertheless, the main obstacle facing the entire area of engineered micro-organisms for degradation is not the inevitable advancement of better technologies but rather the propagation and acceptance of already accomplished successes. Large-scale tests of GEMs have been hampered by public environmental concerns and legislative restrictions, which also impact the standard of fundamental research and, consequently, overall advancement in the field [89,90].

**5. Conclusions**

This review discusses biodegradation as an environmentally friendly and economical approach to the bioremediation of petroleum or petrochemical pollutants in a contaminated environment. The different petroleum products and remediation methods (biological, chemical, and physical-chemical) are presented. Some bioremediation techniques associated with the remediation of petroleum hydrocarbons from contaminated soil and water environments include phytoremediation, bioventing, biophilic, rhizoremediation, biostimulation, and bioaugmentation have been highlighted. These consist of effective petroleum-tolerant plants and microorganisms that can be used in bioremediation. Apart from the microbial community and diversity, some of the environmental and biological influencing variables notable for reducing or favoring degradation productivity in the application of bioremediation were discussed. Research into GEM from different ecologies for the bioremediation of petroleum products is also gaining attention. Therefore, combining bioremediation with GEM and other advanced technologies, such as nanotechnology, to remove contaminants from petroleum hydrocarbons has a promising future.

**Author Contributions:** Conceptualisation E.K.T., J.A.A. and E.K.A.; writing—original draft E.K.T., M.O.A., J.A.A., D.A.-S., E.K.A. and S.O.-F.; writing—reviewing and editing E.K.T., E.K.A. and S.R.; supervision S.R., M.C. and A.H.M.; Administration E.K.A., J.A.A. and M.O.A. All authors have read and agreed to the published version of the manuscript with no conflict of interest.

**Funding:** This research received no external funding.

**Institutional Review Board Statement:** Not applicable.

**Informed Consent Statement:** Not applicable.

**Data Availability Statement:** Not applicable.

**Acknowledgments:** The authors would like to thank the Green Engineering Research Group, under the department of Chemical Engineering Department of the Durban University of Technology, South Africa, The University of KwaZulu-Natal, Durban, South Africa, and Punjabi University, Banda, India, for their support.

**Conflicts of Interest:** The authors state that they have no recognized competing financial or personal interests that might have influenced the work presented in this publication.

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
