# Peer review of "Microbial Bioremediation and Biodegradation of Petroleum Products—A Mini Review"

_applsci, doi:10.3390/app122312212_

Round 1

Reviewer 1 Report

Overall the paper is good and adds to the knowledge but it is still lacking.

A good overview is given on technologies but no detailed discussion of other studies is offered. It is a good summary of the background knowledge but required some critical analysis of other studies' work and the results obtained.

Specifically for section 2/4- I feel that if the authors could add some references in relation to other people finding for the remediation of THP etc it would significantly improve the review.

Section 4- overall gives a good overview but some examples of where their technologies have been used and the result obtained would add to the study.

Reviewer 2 Report

This paper reviewed the bioremediation of petroleum hydrocarbon contaminants in water and soil, focusing on petroleum biodegradable microorganisms essential for the biodegradation of petroleum

contaminants. This review includes enough information, but more critical discussion should be added.

Introduction

1.      Line 88-97, I would like to see some sentences about the current review paper on the same topic and what is the difference between your review and previous review?

2. Petroleum products

1.      The author just listed the types of hydrocarbons degradation and biodegradable pollutants and brief described the characteristics of the pollutants. If you, maybe better to demonstrate this section in a table.

4. Emerging biodegradation technologies

1.      Also in different biodegradation technologies, the author just demonstrated the characteristics. More information such as the example that use this technology and its removal efficiency should be added.

2.      Critical review about different technologies (pros and cons and applications) should be added.

Round 2

Reviewer 2 Report

The author has addressed all my concerns.